# Effects of Physical Properties and Processing Methods on Astragaloside IV and Flavonoids Content in *Astragali radix*

**DOI:** 10.3390/molecules27020575

**Published:** 2022-01-17

**Authors:** Mei-Yin Chien, Chih-Min Yang, Chao-Hsiang Chen

**Affiliations:** 1Ko Da Pharmaceutical Co., Ltd., Taoyuan 324, Taiwan; ss143524@gmail.com (M.-Y.C.); s143524@hotmail.com (C.-M.Y.); 2Graduate Institute of Pharmacognosy, Taipei Medical University, Taipei 110, Taiwan

**Keywords:** *Astragali radix*, astragaloside IV, diameter, flavonoids, processing, turnover rate

## Abstract

The aim of this study was to investigate the effects of the physical properties (diameter size, powder particle size, composition of bark- and wood-tissue, and turnover rate) and processing methods on the content of active ingredients in *Astragali radix* (AR), a popular Chinese herbal medicine. The astragaloside IV and flavonoid contents increased with decreasing diameter size. Bark-tissue had significantly higher astragaloside IV and formononetin content than that in the wood-tissue. As a higher proportion of bark-tissue is associated with decreasing diameter, a strong correlation was also shown between bark- to wood-tissue ratio and active ingredients’ content. Furthermore, an increase in astragaloside IV content was observed in thin powder as compared to coarse powder ground from the whole root. However, this association between active ingredients’ content and powder particle size was abolished when isolating bark- and wood-tissue individually. Moreover, AR stir-frying with refined honey, a typical processing method of AR, increased formononetin content. The turnover rate of active constituents upon decoction ranged from 61.9–81.4%. Assessing the active constituent contents using its physical properties and processing methods allows for a more comprehensive understanding of optimizing and strengthening the therapeutic potentials of AR used in food and herbal supplements.

## 1. Introduction

*Astragali radix* (AR), the dried root of *Astragalus membranaceus* (Fisch.) Bge. var. *mongholicus* (Bge.) Hsiao, or *Astragalus membranaceus* (Fisch.) Bge. (Fam. Leguminosae) and also known as Huangqi in Chinese, is used to treat qi deficiency as well as tonify lung and spleen functions in a traditional way [1]. A large number of active constituents have been isolated from AR, including saponins, flavonoids, isoflavones, glycosides, flavones, polysaccharides, and pterocarpans [2,3]. Several pre-clinical and clinical studies have demonstrated that AR and its active constituents exhibited a broad range of pharmacological effects, including anti-oxidation, immunomodulation; anti-cancer and anti-viral properties; renal, cardiovascular, and hepatoprotective effects [2]. Astragaloside IV and calycosin-7-*O*-β-d-glucopyranoside (Cal-7-O-GLU) are the two active constituents in AR, and their content must meet the acceptance criteria based on the Pharmacopoeia of China (China).

The therapeutic potential of AR and its extensive application in Chinese cuisine classified the herb as a medicine and food homology species. The medicine and food homology concept refers to herbs that can be used both as medicine and food [4]. For culinary purposes, AR is used primarily in soups that are thought to nourish the spleen, enhance energy flow (qi), and boost the immune system. AR is one of the most consumed traditional Chinese herbs. In the Chinese herbal medicine market, the price and grades of AR surge with coarser powder and larger cross-sections [5]. The coarseness and fineness of AR powder are associated with the fibrous content, which is dependent on the composition of bark- and wood-tissue. However, the active ingredient content in different diameter sizes and compositions of AR has not been well understood.

Chinese herbal medicine is a highly personalized treatment. Traditionally, raw materials of Chinese medicinal herbs are processed into decoction pieces in various ways; and these processing methods (paozhi) comprise techniques, such as stir-frying, stir-frying with liquid or solid excipients, steaming, boiling, stewing, etc. Herbs may have a few processing methods and each method potentiates distinct pharmacological benefits and reduces the toxicity of the same herb. The optimal processing method of a specific herb is prescribed to a patient based on the cumulative clinical presentations he or she displayed [6].

In this study, we determined the effects of the physical properties (diameter size, powder coarseness, bark- to wood-tissue ratio) and processing methods (non-processing (NP), stir-frying (SF), stir-frying with rice wine (SFRW), and stir-frying with refined honey (SFRH)) on the active ingredient contents of AR. We further determined the turnover rate of the active ingredient upon decoction, and these results provide insights into maximizing the therapeutic effects of AR.

## 2. Results and Discussion

### 2.1. Effect of Diameter Size on the Content of Astragaloside IV and Flavonoids

While price and grade of AR are positively correlated with the diameter, we sought to determine the effects of diameter size on the content of astragaloside IV and flavonoids in AR. The content of astragaloside IV increased with decreasing diameters (Figure 1). Similarly, Cao et al. [7] also demonstrated that the grade of AR increased with increased diameter size, but the astragaloside IV content was shown to be the lowest in the highest-grade sample in 87 samples that were collected from different origins spanning various grades with the diameter size ranging between 0.7–1.9 cm.

Subsequently, we compared the content of astragaloside IV between extraction methods suggested by the China and the Hong Kong Chinese Materia Medica Standards (HK). Using the China and the HK extraction methods, astragaloside IV content was 0.617% and 0.214% in <0.3 cm subgroup, and 0.146% and 0.035% in 0.9–1.2 cm subgroup, respectively. The content of astragaloside IV decreased by 4.2- and 6.1-fold comparing the two subgroups using the extraction method documented in the China and the HK, respectively (Figure 1). In addition, the content of astragaloside IV was much higher when using the extraction method in the China than that of the HK, and the difference was around 2 to 3-fold (Figure 1). An astragaloside IV content in all subgroups using the China method reached the threshold of no less than the 0.08% established by China [8]. However, astragaloside IV content in the subgroup of 0.9–1.2 cm using the HK extraction method did not achieve its own acceptance criterion of any less than 0.04%. Heating is the major difference between the two extraction methods. Dai et al. [9] demonstrated that astragaloside I, isoastragaloside I, and isoastragaloside II were unstable under heating conditions and these astragalosides may be transformed into astragaloside IV during heating. Therefore, heating, a principal factor increasing the deacetylation reaction, could cause the transformation of astragalosides [9]. In addition, astragaloside IV loss may be attributable to the complicated steps involved in the HK extraction method.

Furthermore, we explored the effect of diameter size on flavonoids content in AR. We identified flavonoids using HPLC and showed that Cal-7-O-GLU, Formononetin-7-O-glucoside (For-7-O-GLU), formononetin, unknown 1 (retention time at 9.3 min), and unknown 2 (retention time at 15.1 min) were well-differentiated on the HPLC chromatogram in all subgroups (Figure 2A). Similar to astragaloside IV, we discovered that flavonoid content decreased with increasing diameter size (Figure 2). In contrast to our results, Liu et al. [10] found that no correlation existed between the content of calycosin glucoside and a diameter size up to 1.9 cm in 52 AR samples collected from different sources spanning across diverse grades. Sun et al. [11] also demonstrated that no correlation was shown between flavonoid content and a diameter ranging between 0.6–2.2 cm in 18 samples of different grades collected from Shanxi province in China.

Based on the quantitative data, Cal-7-O-GLU was the most abundant flavonoid, which accounted for 50% of total flavonoids (Figure 2B). The difference in flavonoid content between subgroups (<0.3 cm and 0.9–1.2 cm) was around 1.5 to 2.5-fold (Figure 2B). Similar findings were observed in a previous study that the content of Cal-7-O-GLU was the highest content among five flavonoids in AR collected from different sources [12]. Notably, Cal-7-O-GLU content in the subgroups of 0.6–0.9 cm and 0.9–1.2 cm did not reach the acceptance criterion of not less than 0.02% set by the China. Taken together, astragaloside IV and flavonoids increase with a decreasing diameter size of AR, and the China method preserves more astragaloside IV than the HK method. While we suggested a diameter-dependent effect on active ingredient contents, several other factors, including sources, age, growth mode (wild, semi-wild, bionic wild, or transplanting), seasonal variation, and species, could also affect active ingredient contents in AR [13,14,15,16,17,18,19,20,21,22].

### 2.2. Effects of Powder Particle Size on the Content of Astragaloside IV and Flavonoids

Astragaloside IV content in the thin powder subgroup was significantly higher than that in the coarse powder subgroup and the difference was 1.7 and 2.8-fold when using the China and HK extraction methods, respectively (Figure 3). This suggested that astragaloside IV content was affected by powder particle size. A possible explanation is that smaller particle sizes exhibited larger surface area, leading to having higher extraction efficiency. Moreover, astragaloside IV content in subgroups of coarse powder and thin powder exhibited statistical differences as compared to the proportional allocation subgroup (Figure 3). The theoretical value was calculated based on the ratio of the coarse and thin powder (2:8, *w*/*w*). Astragaloside IV content in the proportional allocation subgroup exhibited no statistical difference from the theoretical value (Figure 3), implicating that proportional allocation truly reflects astragaloside IV content to avoid misjudgment. The same procedure was carried out to determine whether flavonoids content is dependent on powder particle size. However, no significant difference was observed between flavonoids content and powder particle size (Figure 3). Further exploration is needed to illustrate why flavonoids content was not affected by powder particle size. In brief, the best approach for determining active ingredients’ content was through proportional allocation according to its powder particle size.

### 2.3. Effects of the Bark- to Wood-Tissue Ratio on the Content of Astragaloside IV and Flavonoids

The content of astragaloside IV in bark-tissue was significantly higher than that in the wood-tissue and the difference was around 2 to 3-fold using the China and HK extraction methods. (Figure 4A). Consistent with our results, astragalosides have been reported to be distributed mainly in the bark-tissue of AR [23]. Similarly, astragaloside IV content in all subgroups obtained from the China extraction method was much higher than that of the HK extraction method (Figure 4A). However, no statistical difference was found in astragaloside IV content among all subgroups in bark- and wood-tissue (Figure 4A), indicating that powder particle size in bark- and wood-tissue was not the primary factor affecting the content of astragaloside IV. Moreso, flavonoids content in all subgroups of bark-tissue exhibited no statistical difference and similar findings could be observed in all subgroups of wood-tissue (Figure 4B). Only formononetin and total flavonoids had a significantly higher content in all subgroups of bark-tissue than that of the wood-tissue (Figure 4B).

As a larger diameter size of AR indicates a smaller surface area of bark- to wood-tissue ratio, the diameter-dependent effect of the active ingredients’ content can be further explored using the composition. As shown in Figure 5, the surface area ratio of bark- to wood-tissue increased with decreasing diameter. Zhang et al. [17] have indicated that the age of AR played a more determinant role in the content of active components than that of geographical location. Herein, we also observed that the smaller diameter size was mainly distributed among branch roots belonging to the younger roots, and the proportion of bark-tissue in the branch roots was more than that of wood-tissue. The computed correlation coefficient, r value between the surface area ratio of bark- to wood-tissue and the active ingredient contents (astragaloside IV and flavonoids) were higher than 0.90, indicating that the two were highly correlated (Table 1). However, previously we only observed a dependent relationship between diameter size and astragaloside IV content but no significant difference in the content of Cal-7-O-GLU, and For-7-O-CLU was observed in bark- and wood-tissue (Figure 4B). Therefore, the correlation in certain flavonoids, such as Cal-7-O-GLU and For-7-O-CLU, content between diameter size and the proportion of bark-tissue is doubtful, with further exploration needed. Taken together, only astragaloside IV content could be affected by powder particle size in the whole root of AR (Figure 3) but not in bark- and wood-tissue (Figure 4A). We speculated that powder particle size was not the chief factor, but the proportion of bark-tissue seemed to play a crucial role in the content of astragaloside IV.

### 2.4. Effects of Different Processing Methods on the Content of Astragaloside IV and Flavonoids

Processing methods of Chinese herbal medicine play a critical role in strengthening pharmacological effects [24]. The traditional Chinese medicine theory not only described the use of each herb but also its respective processing methods when treating different symptoms. For example, non-processed or stir-frying with refined honey are the two typical processing methods of AR and the latter was given to patients that also depict weak spleen and stomach absorption in addition to qi-deficiency; honey also boosts the immune system, moisturizes the large intestine, and facilitates defecation. The sophistication and complexity of Chinese medicine theory have been a valuable part of the Chinese heritage. However, in the last few decades, the traditional Chinese medicine industry has evolved into a scientific discipline that requires the validation of efficacy and safety through scientific approaches.

Active constituent content was determined using various processing methods, namely, NP, SF, SFRW, and SFRH. To ensure the validity of the results, the diameter size of 0.6–0.9 cm used for all groups was controlled. The astragaloside IV content in the SF and SFRH subgroups slightly but significantly decreased as compared to the NP subgroup using the China extraction method; however, no statistical difference was shown among all four subgroups using the HK extraction method (Figure 6A). The content of Cal-7-O-GLU and For-7-O-GLU in the SF subgroup reduced 25% and 37%, as compared to the NP subgroup, respectively (Figure 6B). Formononetin content in SF, SFRW, and SFRH subgroups increased by 1.7 to 1.8-fold, as compared to NP subgroup (Figure 6B). The total flavonoids content in the SFRW and SFRH subgroups showed a significant increase by 1.3-fold, as compared to the NP subgroup (Figure 6B). Song et al. [25] have compared the effects of different processing methods on the content of flavonoid compounds and found that the content of Cal-7-O-GLU, For-7-O-GLU, calycosin, and formononetin processed by SFRH showed a statistically significant decrease; however, calycosin content processed by SFRW increased as compared to NP. Shi et al. [26] also determined astragaloside IV and Cal-7-O-GLU content in AR that were collected from different sources using the SFHR method and found that the content of these active compounds displayed a significant decrease in SFHR samples as compared to NP samples. A possible explanation for the difference in active constitutes content in AR between non-processed and processed subgroups is that flavonoids were unstable at higher temperatures and could transform into the aglycones form during the processing procedures [26]. Dai et al. [9] have investigated the thermal stability of isoflavonoids in the water decoction and thus found that the content of calycosin 7-*O*-β-d-(6”-malony)-glucoside and 6”-O-malonylononin may degrade into Cal-7-O-GLU and For-7-O-GLU during the heating condition, further studies are needed to illustrate the relationship between the changes of active components content and the underlying transformation mechanisms in different processing methods.

### 2.5. Turnover Rate of Astragaloside IV and Flavonoids upon Decoction

During the manufacturing process of concentrated extract products, raw materials of Chinese herbal medicine were decocted, filtrated, and concentrated; the loss of active components in the final product is uncertain. While some practitioners still follow the traditional approach, most have chosen to use a concentrated extract powder due to convenience and standardized quality and safety of the product. Decoction is a preferable method for extracting oils and other chemical compounds from herbs and has a versatile application in herbal medicine. Herbs are submerged in water and boiled under heating conditions, which may result in the loss of volatile active ingredients. Therefore, there is a need to evaluate the turnover rate of active constituents upon decoction.

The weight of sliced specimens and the yield of lyophilized powder of decoction was 25.10 ± 0.70 g and 9.86 ± 0.48 g, respectively. Astragaloside IV content in sliced specimens using the China and the HK extraction method was 2.40 and 0.81 mg/g, respectively. Additionally, lyophilized powder of decoction contained 3.95 and 1.28 mg/kg of astragaloside IV using the China and HK extraction method, respectively (Table 1). By calculation, the turnover rate of astragaloside IV was 64.5% and 61.9%, respectively, using the China and HK extraction method (Table 1). The difference in the turnover rates of astragaloside IV using the two extraction methods was not statistically significant. In addition to astragaloside IV, we determined the turnover rate of three types of flavonoids. The flavonoids content in the sliced specimens was 0.208, 0.063, and, 0.131 mg/g for Cal-7-O-GLU, For-7-O-GLU, formononetin, and the lyophilized powder of decoction contained 0.341, 0.131, and 0.256 mg/g of Cal-7-O-GLU, For-7-O-GLU, and formononetin, respectively (Table 1). The average turnover rate of Cal-7-O-GLU, For-7-O-GLU, and formononetin was computed to be 64.3%, 81.4%, and 76.4%, respectively (Table 1). The results suggested that the turnover rate of For-7-O-GLU and formononetin was significantly higher than that of astragaloside IV and Cal-7-O-GLU (Table 1).

## 3. Materials and Methods

### 3.1. AR Collection and Preparation of Diverse Test Specimens

AR samples were collected from Gansu province, China, and the source was identified as the dried root of *Astragalus membranaceus* (Fisch.) Bge. var. *mongholicus* (Bge.) Hsiao (Fam. Leguminosae) by our certificated testing center. The voucher specimen was stored properly. The overall preparation of diverse test specimens was summarized in Figure 7. In our routine analysis of active ingredients’ content in Chinese herbal medicine, the test specimens were powdered and sieved by a 40-mesh sifting screen with an aperture of 425 μm. For the preparation of different diameter sizes of samples powder, the whole root was sliced into diameter sizes of <0.3, 0.3–0.6, 0.6–0.9, and 0.9–1.2 cm based on the cross-section size. All sliced samples were powdered and sieved. During the grinding process, around 20% (*w*/*w*) of the whole root sample cannot pass through the 40-mesh sieve due to its fibrous property. The powdered samples, also called proportional allocation samples, were made according to the ratio of thin powder (particle size < 425 μm) and coarse powder (particle size > 425 μm). For the preparation of different powder particle sizes, the diameter size of 0.3–0.6 cm of sliced specimens was selected, powdered, sieved, and divided into three subdivisions, which were coarse powder, thin powder, and proportional allocation.

The bark- and wood-tissue in the whole root were isolated by hand, powdered, sieved, and divided into three subgroups, namely, >425 μm, <425 μm, and proportional allocation. In the proportional allocation subgroup, the proportion of coarse powder and thin powder was around 1:9 for bark-tissue and 3:7 for wood-tissue. The surface area of the two tissues’ different diameter sizes of samples was calculated by the Image J software. The surface area of the bark-tissue was computed by subtracting wood-tissue from the total area. The surface ratio of bark- to wood-tissue was calculated by dividing the surface area of the bark-tissue by the surface area of the wood-tissue.

### 3.2. Preparation of Diverse Paozhi Samples

The diameter size of 0.6–0.9 cm of sliced specimens was selected for the preparation of diverse paozhi samples. The SF samples were prepared by placing 50 g of sliced specimens into a wok and stir-fried over a gentle heat until the surface turn yellow. The SFRW samples were prepared by soaking 50 g of sliced specimens in rice wine at the ratio of 1:5 (*w*/*v*) for 1 h. The soaked samples were subsequently placed into a wok and stir-fried over a gentle heat until the surface turn yellow. The refined honey was prepared by mixing 100 g honey with 400 g water, placed into a wok, and stir-fried until bubbles were formed. The SFRH samples were prepared by soaking 50 g of sliced specimens in prepared refined honey at the ratio of 1:5 (*w*/*v*) for 1 h, placed into a wok, and then stir-fried until the surface turn brown under gentle heating. All prepared samples were dried, powdered, and sieved by passing them through a 40-mesh sifting screen. The powdered samples were made according to the ratio of thin powder (particle size < 425 μm) and coarse powder (particle size > 425 μm).

### 3.3. Preparation of Decoction Lyophilized Powder and Calculation of Turnover Rate

The diameter size of 0.3–0.6 cm of sliced specimens was selected for the preparation of decoction lyophilized powder. An amount of 25 g of sliced specimens were decocted with 250 mL of boiling water twice (1 h for each time). Subsequently, the decoction was filtrated, concentrated, and subjected to a freeze dryer. The weight of decocted specimens and the final lyophilized powder were recorded. The turnover rate was calculated as the following formula: [weight of the lyophilized powder of decoction (g) × active ingredients’ content in lyophilized powder of decoction (mg/g)] ÷ weight of sliced specimens (g) ÷ active ingredients’ content in sliced specimens (mg/g) × 100%.

### 3.4. Determination of the Astragaloside IV Content

Preparation of the test solution according to the monograph of AR documented in the China was performed by precisely placing 1 g of powdered sample and 50 mL of 80% (*v*/*v*) methanol containing 4% (*v*/*v*) ammonium hydroxide into a flask. The flask containing the powdered sample and extraction solvent was weighed and heated under reflux for 1 h. The solution was cooled, weighed again, and 80% (*v*/*v*) methanol containing 4% (*v*/*v*) ammonium hydroxide was added to compensate for the loss of the extraction solvent. After filtration, 25 mL of the filtrate was placed into an evaporating dish and place on a boiler to allow for evaporation to dryness. The residue was dissolved in 10 mL of 80% (*v*/*v*) methanol.

Preparation of the test solution based on the monograph of AR documented in HK was performed by placing 2 g of powdered samples and 30 mL of methanol into a 50 mL centrifugal tube. The mixture was sonicated for 30 min, followed by centrifugation at 3000× *g* for 5 min. The residue in the centrifugal tube was washed with 15 mL of methanol two times and centrifuged at 3000× *g* for 5 min. The supernatant was collected and evaporated to dryness under reduced pressure in a rotary evaporator. The residue was dissolved in 10 mL of 10% (*v*/*v*) ammonium hydroxide and allowed to stand for 10 min with occasional shaking. The dissolved solution was transferred into a separating funnel and extracted with 15 mL of water-saturated 1-butanol. The upper layer solution was collected and the lower layer solution was further extracted with 10 mL of water-saturated 1-butanol three times. The upper layer solution was collected into a flask and evaporated to dryness at reduced pressure in a rotary evaporator. The residue was dissolved in 10 mL of methanol.

The standard solution was prepared by dissolving 2 mg of astragaloside IV into 5 mL of methanol, and serial dilution with methanol was performed to obtain different concentrations. Analytical HPLC was performed on the D-7000 interface (Hitachi, Tokyo, Japan) equipped with the L-7100 pump (Hitachi, Tokyo, Japan), L-7200 auto-sampler (Hitachi, Tokyo, Japan), and 90 LT- evaporative light scattering detectors (ELSD)(SEDERE, Paris, France) to determine astragaloside IV content in various test solutions. Chromatographic separation was carried out on a Mightysil RP-18 column (4.6 × 250 mm, 5 μm) (Kanto, Tokyo, Japan) using a mixture of acetonitrile and water (38:62, *v*/*v*) as the mobile phase. The run time, flow rate, and injection volume were set at 15 min, 1.0 mL/min, and 10 μL, respectively. The ELSD parameters were set at 1.9 L/min for gas flow, 90 °C for drift tube temperature, and 8 for gain value.

### 3.5. Determination of the Flavonoids Content

Preparation of the test solution according to the monograph of AR documented in China was performed by placing precisely 1 g of powdered samples and 50 mL of methanol into a flask. The flask containing powdered sample and extraction solvent was weighed and heated under reflux for 4 h. The sample was then cooled and weighed. Methanol was added to compensate for the loss of the extraction solvent. After filtration, 25 mL of filtrate was evaporated to dryness at reduced pressure in a rotary evaporator. The residue was dissolved in 5 mL of methanol. Preparation of the standard solution was performed by dissolving 2 mg of Cal-7-O-GLU, For-7-O-GLU, and formononetin in 10 mL of methanol. Dilution with methanol was carried out to obtain a different concentration through serial dilution. Analytical HPLC was performed on the Hitachi D-7000 interface equipped with the L-7100 pump, L-7455 detector, and L-7200 auto-sampler (Tokyo, Japan) to determine flavonoid content in test solutions. Chromatographic separation was carried out on a Mightysil RP-18 column (4.6 × 250 mm, 5 μm) (Kanto, Tokyo, Japan) using the gradient solvent system which contained acetonitrile (A) and 0.2% (*v*/*v*) formic acid (B). The gradient profile was set as follows: 0–20 min at 20–40% A; 20–30 min at 40% A; 30–30.1 min at 40–20% A; 30.1–40 min at 20% A. The run time, UV wavelength, flow rate, column temperature, and injection volume were set as 40 min, 260 nm, 1.0 mL/mL, 40 °C, and 10 μL, respectively.

### 3.6. Statistical Analysis

Numerical variables were expressed as mean ± SD. Analysis was carried out using the one-way ANOVA followed by Fisher’s least square difference test in making pairwise comparisons of group means. The statistical analysis was performed using SPSS Statistics Version 22 (SPSS, Inc., Chicago, IL, USA). A *p* value of less than 0.05 was considered statistically significant.

## 4. Conclusions

In summary, the present study identified the bark-tissue proportion and smaller diameter size of AR as important indicators of higher astragaloside IV content, as evidenced by (1) an increasing astragaloside IV content with decreasing diameter size but increasing bark- to wood-tissue ratio; (2) the negative correlation between diameter size and bark- to wood-tissue surface area ratio; and (3) a strong correlation between astragaloside IV content and bark proportion. Moreover, we determined that stir-frying with rice wine and refined honey increases formononetin content as compared to non-processed AR samples. The turnover rate of active constituents in AR upon decoction ranged from 61.9% to 81.4%. Taken together, these findings provide insights into key factors affecting active ingredients in AR and this allows for the optimization of therapeutic efficacy for consumers, practitioners, and manufacturers.

## Figures and Tables

**Figure 1 molecules-27-00575-f001:**
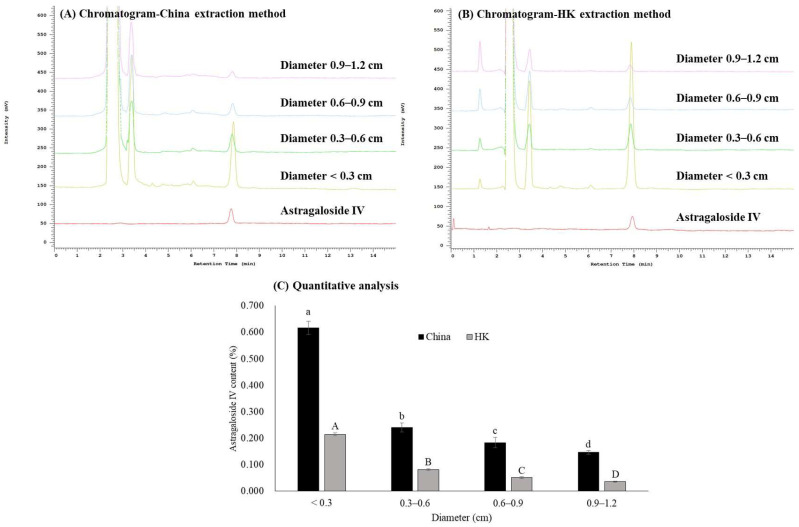
Effects of different diameters on the content of Astragaloside IV in *Astragali radix* (AR). HPLC chromatogram of standard solution (Astragaloside IV) and test solution (AR with different diameter sizes) based on the China (**A**) and HK (**B**) extraction method. (**C**) Quantitative analysis of Astragaloside IV content with different diameter sizes. Data are expressed as mean ± SD from three separate experiments; means not sharing an alphabetic letter (lowercase for China and uppercase for HK) differs statistically (*p* < 0.05). Abbreviations (alphabetically): China, the Pharmacopoeia of China; HK, the Hong Kong Chinese Materia Medica Standards.

**Figure 2 molecules-27-00575-f002:**
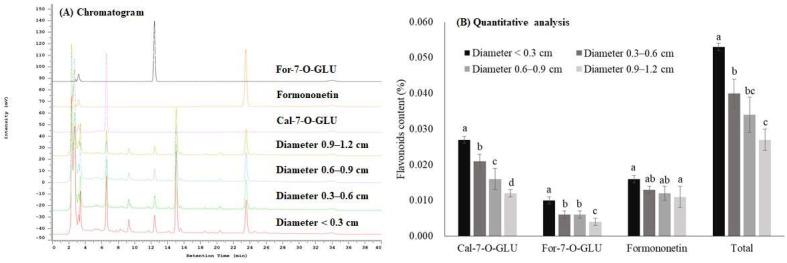
Effects of different diameters on the content of flavonoids in *Astragali radix* (AR). (**A**) HPLC chromatogram of standard solution (individual flavonoid) and test solution (different diameter sizes). (**B**) Quantitative analysis of flavonoids content with different diameter sizes. Data are expressed as mean ± SD from three separate experiments; means not sharing an alphabetic letter (lowercase for Cal-7-O-GLU, uppercase for For-7-O-GLU, italic lowercase for formononetin, and italic uppercase for total flavonoids) differ statistically (*p* < 0.05). Total flavonoids is the sum of Cal-7-O-GLU, For-7-O-GLU, and formononetin. Abbreviations (alphabetically): Cal-7-O-GLU, Calycosin-7-*O*-β-d-glucopyranoside; For-7-O-GLU, Formononetin-7-O-glucoside.

**Figure 3 molecules-27-00575-f003:**
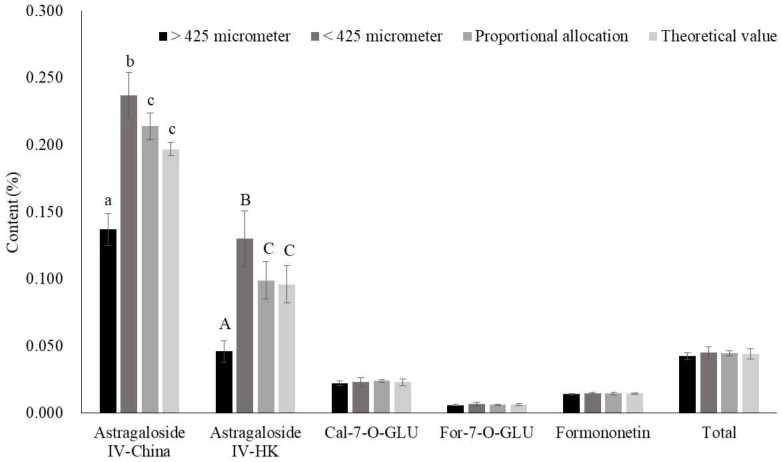
Effects of powder particle size on the content of Astragaloside IV and flavonoids in *Astragali radix* with a diameter of 0.3–0.6 cm. Data are expressed as mean ± SD from three separate experiments; means not sharing an alphabetic letter (lowercase for China and uppercase for HK) differs statistically (*p* < 0.05). Total flavonoids is the sum of Cal-7-O-GLU, For-7-O-GLU, and formononetin. Abbreviations (alphabetically): Cal-7-O-GLU, Calycosin-7-*O*-β-d-glucopyranoside; China, the Pharmacopoeia of China; For-7-O-GLU, Formononetin-7-O-glucoside; HK, the Hong Kong Chinese Materia Medica Standards.

**Figure 4 molecules-27-00575-f004:**
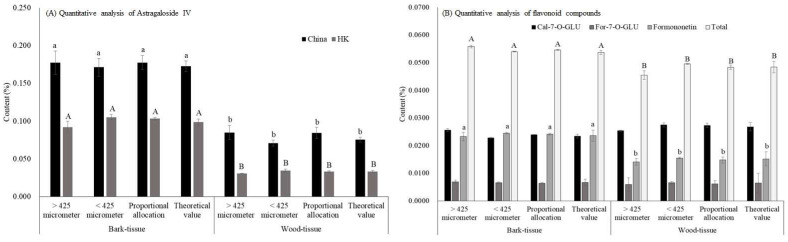
Effects of bark- and wood-tissue with diverse powder particle size on the content of Astragaloside IV (**A**) and flavonoids (**B**) in *Astragali radix*. Data are expressed as mean ± SD from three separate experiments; means not sharing an alphabetic letter (left panel: lowercase for China and uppercase for HK; right panel: lowercase for formononetin and uppercase for total flavonoids) differ statistically (*p* < 0.05). Total flavonoids is the sum of Cal-7-O-GLU, For-7-O-GLU, and formononetin. Abbreviations (alphabetically): Cal-7-O-GLU, Calycosin-7-*O*-β-d-glucopyranoside; China, the Pharmacopoeia of China; For-7-O-GLU, Formononetin-7-O-glucoside; HK, the Hong Kong Chinese Materia Medica Standards.

**Figure 5 molecules-27-00575-f005:**
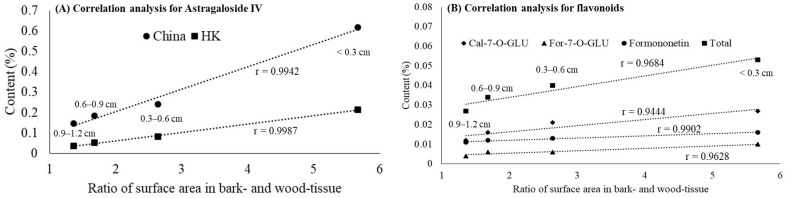
Correlation analysis between the active component content (Astragaloside IV and flavonoids) and the ratio of surface area in bark- and wood-tissue surface area in *Astragali radix* with different diameters. The surface area of bark and wood-tissue was calculated by Image J software and the surface area of bark was computed by subtracting wood-tissue from the total area. The ratio of surface area in bark and wood-tissue was calculated as bark area divided by wood-tissue area. Total flavonoids is the sum of Cal-7-O-GLU, For-7-O-GLU, and formononetin. Abbreviations (alphabetically): Cal-7-O-GLU, Calycosin-7-*O*-β-d-glucopyranoside; China, the Pharmacopoeia of China; For-7-O-GLU, Formononetin-7-O-glucoside; HK, the Hong Kong Chinese Materia Medica Standards.

**Figure 6 molecules-27-00575-f006:**
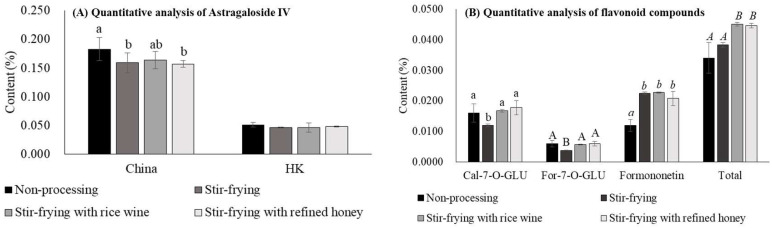
Effects of diverse processing methods on the content of Astragaloside IV (**A**) and flavonoids (**B**) in *Astragali radix* with a diameter of 0.6–0.9 cm. Data are expressed as mean ± SD from three separate experiments; means not sharing an alphabetic letter (left panel: lowercase for China; right panel: lowercase for Cal-7-O-GLU, uppercase for For-7-O-GLU, italic lowercase for formononetin, and italic uppercase for total flavonoids) differ statistically (*p* < 0.05). Total flavonoids is the sum of Cal-7-O-GLU, For-7-O-GLU, and formononetin. Abbreviations (alphabetically): Cal-7-O-GLU, Calycosin-7-*O*-β-d-glucopyranoside; China, the Pharmacopoeia of China; For-7-O-GLU, Formononetin-7-O-glucoside; HK, the Hong Kong Chinese Materia Medica Standards.

**Figure 7 molecules-27-00575-f007:**
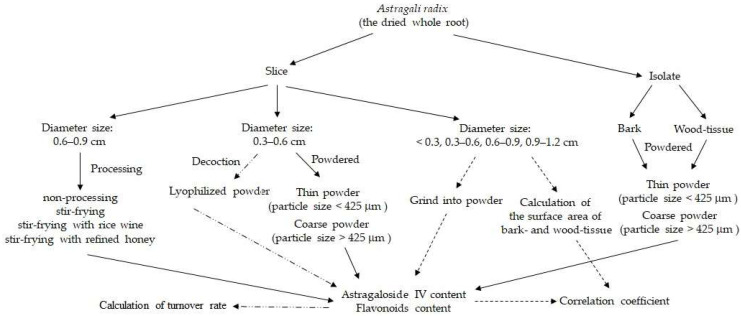
The overall preparation of diverse test specimens.

**Table 1 molecules-27-00575-t001:** Turnover rate of Astragaloside IV and flavonoids from raw materials to decoction in AR with a diameter of 0.3–0.6 cm.

Items	Astragaloside IV		Flavonoids		
	China	HK	Cal-7-O-GLU	For-7-O-GLU	Formononetin
Content in slice specimens (mg/g)	2.40 ± 0.17	0.81 ± 0.03	0.208 ± 0.025	0.063 ± 0.007	0.131 ± 0.010
Content in lyophilized powder of decoction (mg/g)	3.95 ± 0.34	1.28 ± 0.07	0.341 ± 0.029	0.131 ± 0.005	0.256 ± 0.038
Turnover rate (%)	64.5 ± 3.8 ^a^	61.9 ± 3.6 ^a^	64.3 ± 3.2 ^a^	81.4 ± 1.4 ^b^	76.4 ± 9.0 ^b^

Data were expressed as mean ± SD from three independent experiments; means not sharing an alphabetic letter differ statistically in turnover rate (*p* < 0.05). Abbreviations (alphabetically): Cal-7-O-GLU, Calycosin-7-*O*-β-d-glucopyranoside; China, the Pharmacopoeia of China; For-7-O-GLU, Formononetin-7-O-glucoside; HK, the Hong Kong Chinese Materia Medica Standards.

## Data Availability

Data sharing is not applicable to this article.

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
