# Peer review of "Effects of Physical Properties and Processing Methods on Astragaloside IV and Flavonoids Content in Astragali radix"

_molecules, 2022, doi:10.3390/molecules27020575_

Round 1
Reviewer 1 Report
The manuscript needs some revision before it can be considered further.
1. A response surface method analysis should be included in the prioritization part because only four factors have been considered in the processing procedure.
2. Error bars should be added in Fig 5.
3. A discussion of the impact on changes of metabolic profiles by the application of different processing methods should be made in detail.
Reviewer 2 Report
Dear authors,
you will find my comments/suggestions below this message.
REPORT
NOVELTY AND IMPACT
The manuscript intends to provide scientific evidence to the general practice of relating coarser powder and larger cross-sections with superior AR grade and therefore, higher astragaloside IV and flavonoids content.
GRAMMAR AND FORMAT
Language editing is recommended. Please, check throughout the text that Astragali radix should be written in italics as well as the name of the plant species.
Journal names should be written in italics as recommended by the ACS style guide.
CONTENT
INTRODUCTION
Line 46: it should say “wood-tissue”.
- When the authors refer to the herbal medicine, it is important to explain if the whole root is used, or if it is the bark or the wood.
- The full botanical name including family should be provided.
- RESULTS AND DISCUSSION
2.1 Effect of diameter size of AR on the content of astragaloside IV and flavonoids
- Lines 65-67: It is not clear what the authors mean when they refer to the “whole plant”. According to Materials and Methods, they only employed the dried roots so the expression “whole plant” should be corrected. The information provided here is different from the one stated in 3.1. AR collection and preparation. Please, clarify.
- Line 78: according to Kafle et al. 2021, the recent edition of Chinese pharmacopoeia 2020 stated, that the threshold of AG-IV must not be less than 0.08% (w/w). Please, verify.
- Lines 79-83: the paragraph should be rewritten.
- Line 93: the authors state that “we validated flavonoids using HPLC” This expression is incorrect. They might have identified flavonoids, not validated them.
- Lines 97-102: This paragraph should be rewritten. It is not clear which findings belong to the authors and which to other researchers. Also, grammar should be checked: “similar findings was observed” is a mistake. It should be “were observed”.
- Line 105: same mistake, “similar findings was observed”
- Line 109: “Taken together, astragaloside IV and flavonoids increases with decreasing diameter size…” It should say “increase”
- The authors should have considered sample diameter size and plant age as well. It could be hypothesized that thinner roots are younger and so flavonoid content might be higher.
2.2 Effects of powder particle size on the content of astragaloside IV and flavonoids in AR
- Lines 125-130
- The information regarding powder sieving should be explained in the Materials and Methods section.
- The use of the term “pro-rata” to refer to the third subdivision is confusing.
- Why was the 0.3-0-6 cm sample the only one that was grinded and used for this test?
- Line 133: the authors state that “This suggested that astragaloside IV content was affected by powder particle size”. What is a possible explanation for this finding? Flavonoids do not seem to be affected by particle size.
2.3 Effects of the bark- to wood-tissue ratio on the content of astragaloside IV and flavonoids in AR
- Lines 151-153
- The information regarding powder subdivision should be placed in the Materials and Methods section.
- There is no need to write AR every single time. It is clear that the authors worked with this plant material.
- At the beginning of the paragraph “Bark- and wood-tissue of AR was isolated” is a mistake. It should say “were isolated”.
- Line 176 and following
- The term “validated” should be removed.
- The explanation provided to show the possible correlation between content and particle size is rather confusing and should be improved because at times it seems contradictory. From figure 5 and general experience, it would be logical to find a positive correlation between smaller particle size and content mainly because smaller particles exhibit larger surface area and so extraction efficiency is enhanced.
- Another thing to consider is the plant or root age. If smaller diameter size corresponds to young roots, bark tissue is expected to be more developed than wood.
2.4 Effects of different processing methods on the content of astragaloside IV and flavonoids in AR
- Lines 213-215
- What is the rational for choosing 0.6-0.9 cm diameter size?
- Not only processing methods could influence compound stability but also the extraction method. ChP uses drastic extraction conditions compared to HK and could provide an explanation as to why no significantly statistically difference was found when HK method was employed. A stability study should be performed. In addition, it would be interesting to consider the discussion provided by Kafle et al. 2021 about the addition of ammonia for extraction purposes, mentioned in the ChP.
2.5 Turnover rate of astragaloside IV and flavonoids upon decoction
- Lines 257-260
- The phrase “While the resulting decoction liquid is a great method of drug delivery due to the prompt absorption of nutrients..” should be changed. A decoction cannot be considered a drug delivery method with enhanced absorption simply because there are many factors to be considered such as compound nature (aglycone, glycoside derivative or other), bioavailability, solubility, stability, transport to the target, and so forth.
- Table 1 should be changed. The weight of raw material and lyophilized powder should be placed somewhere else. The turnover calculation should be included in the Materials and Methods section in an equation form, including units. The way in which the formula is written at the bottom of the table is very confusing.
- Line 273
- Table 2 is missing.
- MATERIALS AND METHODS
As a general comment, the methodological part should be revised and improved for clarification purposes.
3.1 AR collection and preparation: this whole section should be revised.
- Lines 287-296: This paragraph should be revised.
- The explanation provided here is different from the one stated in 2.1. It seems that the authors 1) divided the samples (in this case, the dried roots not the whole plant) into 4 subgroups based on the root diameter size (< 0.3 cm; 0.3-0-6 cm; 0-6-0.9 cm; 0.9-1.2 cm), 2) they obtained the cross sections for each group. If this is the case, the paragraph should be completely rewritten because in its present form it is not clear.
- Lines 298 and following
- What is the rationale for choosing only AR samples of 0.6-0.9 cm for stir-frying? Were the slices placed in the wok or the whole sample? This should be explained.
- Why was the decoction and subsequent lyophilized samples prepared from the 0.3 – 0.6 cm subgroup exclusively?
3.2 Determination of the astragaloside IV content
- Lines 313 and following
- The authors refer to the ChP monograph and they work with a powder. How was this obtained? Which samples were processed? Does the term “sample” refer to the dried root or a processed sample?
- Lines 316- 317: it should say “weighed” not “weighted”.
- There is a significant difference between the two pharmacopeial methods, especially for the use of high temperatures in the ChP method. Has compound stability been verified with such drastic conditions?
- Lines 335 – 345
- The authors should check units, a micro symbol (µ) is missing in the column particle size and the injection volume.
- It should say run time, not analytical time.
3.3 Determination of the flavonoids content
- Lines 347 – 364
- The authors should check units, a micro symbol (µ) is missing in the column particle size and the injection volume.
- The powder that is used as a sample, how was it obtained? Which is exactly the sample in this case? This needs to be clarified.
- Again, the extraction conditions for flavonoids are very drastic (4-hour-reflux). Has flavonoid stability been checked?
- Gradient conditions: acetonitrile (A) and 0.2 % (v/v) formic acid. The letter B is missing
- The gradient conditions are usually expressed with reference to either A or B but not both. For example: 0-20 min 20-40% A; 20-30 min 40% B.
- Check this part, something is not right: 30-30.1 min with the ratio of 40-20% A and 60-80% B; 30-40 min (it should be 30.1 – 40 min) with the ratio of 20% A and 80% B.
- What was the run time?
- When the authors refer to total flavonoids, is this the sum of the three individual flavonoids? Please, clarify.
FIGURES AND TABLES
- Figure 2 legend: there is a mistake in the compound name Cal-7-O-GLU. It cannot be glucopynoside. Also, a symbol is missing in the sugar configuration.
- TABLE 2 is missing
- It would be highly advisable to provide an additional figure to show the sample process flow and the tests performed at each stage.
CONCLUSIONS
- The authors state that formononetin content increases with processing. What could be a possible explanation for this? In addition, further explanation should be provided as to why decreasing diameter size seems to affect astragaloside IV content but not flavonoids’.
REFERENCES
Some additional references could be considered to update and enrich the discussion
- Dai et al. 2020. Quality Markers for Astragali Radix and Its Products Based on Process Analysis. The authors studied compound stability.
- Kafle et al. 2021. Major bioactive chemical compounds in Astragali Radix samples from different vendors vary greatly.
DECISION
I consider that the manuscript needs a major revision as per the suggested observations.
Round 2
Reviewer 2 Report
Dear authors,
Just some minor points to correct
- Lines 69-73: Please, check syntax because when it says “87 samples were collected” it should say “87 samples collected” or “that were collected”.
- Lines 271 – 274: Please, check sentence syntax. It seems that the word “although” should be deleted